# Gender-Related Outcomes after Surgical Resection and Level of Satisfaction in Patients with Left Atrial Tumors

**DOI:** 10.3390/jcm12052075

**Published:** 2023-03-06

**Authors:** Viyan Sido, Annika Volkwein, Martin Hartrumpf, Christian Braun, Ralf-Uwe Kühnel, Roya Ostovar, Filip Schröter, Sofia Chopsonidou, Johannes Maximilian Albes

**Affiliations:** Department of Cardiovascular Surgery, Heart Center Brandenburg, University Hospital Brandenburg Medical School, Faculty of Health Sciences Brandenburg, 16321 Bernau bei Berlin, Germany

**Keywords:** cardiac tumor, atrial myxoma, gender, embolism

## Abstract

Background: Cardiac tumors are rare, with a low incidence of between 0.0017 and 0.19%. The majority of cardiac tumors are benign and predominantly occur in females. The aim of our study was to examine how outcomes differ between men and women. Methods: From 2015 to 2022, 80 patients diagnosed with suspected myxoma were operated on. In all patients, preoperative, perioperative, and postoperative data were recorded. Such patients were identified and included in a retrospective analysis focused on gender-related differences. Results: Patients were predominantly female (*n* = 64; 80%). The mean age was 62.76 ± 13.42 years in female patients and 59.65 ± 15.84 years in male patients (*p* = 0.438). The body mass index (BMI) was comparable in both groups: between 27.36 ± 6.16 in male and 27.09 ± 5.75 (*p* = 0.945) in female patients. Logistic EuroSCORE (LogES) (female: 5.89 ± 4.6; male: 3.95 ± 3.06; *p* = 0.017) and EuroSCORE II (ES II) (female: 2.07 ± 2.1; male: 0.94 ± 0.45; *p* = 0.043), both scores to predict the mortality in cardiac surgery, were significantly higher in female patients. Two patients died early, within 30 days after surgery: one male and one female patient. Late mortality was defined as the 5-year survival rate, which was 94.8%, and 15-year survival rate, which was 85.3% in our cohort. Causes of death were not related to the primary tumor operation. The follow up showed that satisfaction with surgery and long-term outcome was high. Conclusion: Predominately female patients presented with left atrial tumors over a 17-year period. Relevant gender differences aside from that were not evident. Surgery could be performed with excellent early (within 30 days after surgery) and late results (follow up after discharge).

## 1. Introduction

Left atrial myxomas are rare, but they are the most common form of benign cardiac tumors. However, they are seen quite frequently in cardiac surgery facilities. They can affect any chamber of the heart. The incidence of myxomas ranges from 0.0017 to 0.19% [1]. Various studies show a prevalence of almost 0.03% in the general population [2]. These cardiac tumors occur more frequently in women than in men. Some studies report a 3:1 ratio between the two sexes [3], while other studies show that 53–77.4% of myxoma patients are female [4]. The tumor is usually located in the left atrium. About 85% of myxomas are found there [5,6]. In a few cases, they occur in both atria, or in the ventricles. The tumor can be seen in all age groups [6]. However, the peak age of patients with myxomas is 56 years [6]. There are also other tumor entities. Fibroadenomas are less often found in the left atrium. They are more commonly located on the aortic valve but can also be found on the mitral valve and are therefore not easily distinguished from a myxoma with diagnostic imaging techniques such as CT, MRI, or echocardiography. Fibroelastomas account for 7.9% of benign cardiac tumors [7]. Occasionally, suspected tumors of the left atrium turn out to be mere thrombi. Since even these cannot be clearly distinguished from a myxoma or fibroadenoma, definitive proof can only be provided by surgery and histopathological examination. Malignant tumors of the heart are extraordinarily rare. Among them, sarcomas are found most often [8]. They can present as a benign tumor and be misinterpreted in such a way that surgery is performed, although the prognosis is poor, and surgery might not have been indicated given the poor prognosis and thus the fruitlessness of the operation. Both location and tissue structure can lead to serious complications, such as direct embolic or thromboembolic events affecting, for example, the central nervous system [9], or they can affect inflow and outflow through the respective chamber or even directly obstruct valve function. They can thus cause a plethora of clinical symptoms such as palpitations, elusive neurological disorders, or other organ insufficiencies such as renal failure or symptoms of heart failure. However, they are often discovered incidentally [10]. The diagnostic approach in patients with a cardiac tumor is based on history, clinical examination, and especially, echocardiography. Figure 1 shows an exemplary CT- scan (a) and echocardiographic findings (b, c) of a left atrial myxoma attached to the interatrial septum. To date, there have not been many comparative studies on the topic of gender differences in these rare cardiac tumors. The aim of this retrospective study was to investigate these differences in rare diseases such as myxoma.

## 2. Materials and Methods

We retrospectively analyzed the medical records of 80 patients who underwent operative resection of suspected cardiac myxoma from 2005 and 2022 in our heart center. Their clinical data were analyzed and compared between both genders (male and female). All patients underwent surgical removal of the left atrial tumor. The tumors were resected by median sternotomy, cardiopulmonary bypass, and X-clamping with the use of cardioplegia. Intraoperatively performed echocardiography was used to proof complete eradication of the tumor. Epidemiological data, risk factors, co-morbidities, and periprocedural data were analyzed and compared between men and women. Furthermore, concomitant interventions, surgery times such as the duration of the surgery, or x clamp and bypass times were recorded, as well as preoperative symptoms of the patients. Histological findings were also determined. For tables and figures, as well as graphic and statistical evaluation, we used Microsoft Excel (Microsoft, Redmond, WA, USA), SPSS (IBM, Armonk, NY, USA) and R (R Core Team, Vienna, Austria) [11]. All data were expressed as continuous or categorical variables. The continuous numerical values were expressed by means ± standard deviation, and the categorical variables were expressed in percentages. Numerical data was compared using Student’s *t*-test if normally distributed and Mann–Whitney U test otherwise. Categorical data was compared using Fisher’s exact test and chi^2^ test, respectively. A value of *p* < 0.05 was considered significant. Long-term survival was obtained after 5 years and 15 years, and survival probability was obtained with Kaplan–Meier survival curves created using the survival package in R [12].

## 3. Results

The number of male patients (*n* = 16; 20%) and female patients (*n* = 64; 80%) differed markedly in our study group. The mean age was 62.76 ± 13.42 years in female patients and 59.65 ± 15.84 years in all male patients and did not differ statistically (*p* = 0.438). Body mass index (BMI) was comparable in both groups presenting between 27.36 kg/m^2^ ± 6.16 in male and 27.09 ± 5.75 kg/m^2^ in female patients (*p* = 0.945).

Preoperative risk scores such as LogES or EuroSCORE II, which predict mortality in cardiac surgery, were higher in women than in men (EuroSCORE: female patients: 5.42 ± 2.41 vs. male patients: 4.04 ± 2.14, *p* = 0.013). The Logistic EuroSCORE (logES) as well as the EuroSCORE II were also found to be higher for women than for men (LogES: female patients: 5.89 ± 4.6 vs. male patients: 3.95 ± 3.06, *p* = 0.017; EuroSCORE II: female patients: 2.07 ± 2.1 vs. male patients: 0.94 ± 0.45, *p* = 0.043) (Figure 2).

The cross-clamp time (female patients: 41.62 ± 29.61 min vs. male patients: 41.8 ± 37.8 min, *p* = 0.6) and cardiopulmonary bypass time (female patients: 77.45 ± 40.86 min vs. male patients: 71.73 ± 45.28 min, *p* = 0.24) showed no significant difference in both genders (Table 1).

The most reported symptoms on admissions were dyspnea (female patients: 25.49% vs. male patients: 16%, *p* = 0.398) and atrial fibrillation (female patients: 27.45% vs. male patients: 19.23%, *p* = 0.609) (see Table 2). Some of the patients reported chest pain (angina pectoris) (female patients 11.76% vs. male patients: 8%, *p* = 1) as a symptom. Comorbidities include pulmonary diseases (female patients: 15.69% vs. male patients: 24%, *p* = 0.573), arterial hypertension (female patients: 70.59% vs. male patients: 61.54%, *p* = 0.586), diabetes mellitus (female patients: 19.61% vs. male patients: 16%, *p* = 1), and hyperlipidemia (female patients: 39.22% vs. male patients: 40%, *p* = 1).

Concomitant coronary atherosclerosis was present in 23.75% (*n* = 19) of all the patients. Eight patients underwent coronary artery bypass grafting (Table 3). Two patients underwent mitral valve surgery, and two others underwent mitral valve repair (Figure 3). Myxoma was histopathologically confirmed in 55 patients (Figure 4; Appendix A). Eleven of all cases showed a thrombus and eight cases a fibroelastoma. Other benign entities were a hamartoma, thyroid gland tissue, and thymus tissue. Three patients showed a malignant tumor: one metastasis of unknown origin, one carcinoid, and one sarcoma (Table 4). Distribution of tumor entities did not differ between male and female patients (Table 4). In-hospital mortality rate was 2.5% (*n* = 2). One patient with sarcoma and one senescent patient (79 years at time of surgery) with a myxoma, who developed a septic shock after treatment of deep sternal wound infection and mediastinitis, died early. No recurrent tumors were identified clinically or by echocardiography in our patients during follow-up.

### Follow-Up Data and Level of Satisfaction

After hospital discharge, follow-up confirmed high life expectancy after myxoma resection: 69 of 80 patients (86.25%) were still alive in 2023, while 11 patients (13.75%) had died. Three patients died from heart disease, one patient from cancer, and two patients from COVID-19 infection. In five patients, the causes of death could not be determined. Three other patients (3.75%) could not be reached by phone, nor could their relatives or their practitioner be reached (Table 5). They were thus lost to follow-up. The 5-year survival was 94.8%, 10-year survival 85.3%, and long-term (15 years) survival was 85.3% (see Figure 5, Kaplan–Meier survival curve).

For follow-up, the patients were called in January 2023 and were asked about their “level of satisfaction”. We wanted to know how well the patients felt cared for during their hospital stay in our clinic. On a scale of 1–5, 1 stood for “very satisfied”, 2 for “satisfied”, 3 for “neutral”, 4 for “unsatisfied”, and 5 for “very unsatisfied” (Table 6).

We also asked the patients about their current level of satisfaction regarding their reintegration into everyday life. Our data showed a reasonable level of satisfaction. During follow up, patients reported a fast recovery and reintegration into everyday life after surgery today (Table 7). Table 6 shows their level of satisfaction in everyday life and during the follow-up, while Table 8 shows the mean and median of the level of satisfaction for both sexes.

## 4. Discussion

The age of our patients varied widely, but most were in their 7th or 8th decade of life, which is consistent with results in the literature [13]. Cardiac myxomas were more common in women than in men in our study, which is also consistent with the findings in the literature. Age in our study ranged from 23 to 83 years.

In most cases, clinical manifestations of cardiac myxoma are nonspecific. Dyspnea is the most frequent symptom, indicating an impaired blood flow predominantly into or through the left atrium [10]. The most frequent occurring symptom in our data was indeed dyspnea, followed by non-specific chest pain or neurological symptoms such as TIA or stroke. Although this kind of cardiac tumor is histologically benign, it can have serious consequences. Thromboembolism or emboli from the tumor itself can cause organ infarction, being particularly serious in the kidney or the brain, where they might lead to stroke [14]. Furthermore, these tumors can have physical consequences, as the tumor can hamper left atrial inflow or proper flow through the mitral valve, resulting in cardiac insufficiency, depending on the specific location. In our series, 23.75% (*n* = 19) of the patients suffered from preoperative embolization such as stroke or transient ischemic attack. Delay of a correct diagnosis can lead to serious complications such as embolization [9]. In our study, TIA and stroke were more frequent in women then in men, but this was not statistically significant. However, one may speculate that a preoperative delay was present in our study and may even be more frequent in women than in men. It can be cautiously speculated that women tend to see a practitioner later than men.

The prevalence of atherosclerotic coronary disease in patients with myxomas ranges from 20.3% to 36.6% [15]. In our study, more men than women suffered from three-vessel coronary disease, which is consistent with the literature on gender differences in the prevalence of coronary artery disease [16,17]. The standard surgical approach is through a median sternotomy, including cardiopulmonary bypass with an aortic and bicaval cannulation. This is also our standard procedure at our heart center for the removal of a myxoma [9]. One may opt for a minimally invasive approach via a right-sided mini-thoracotomy. However, if complications occur, such as dislocation of tumor components into the left ventricle or into the aorta, removal via this approach is not possible or at least very difficult.

According to the WHO, the myxoma belongs to the group of tumors of the pluripotent mesenchyma, although the histogenesis has not been finally clarified [18]. It macroscopically appears to be a polypoid tumor, often with a round to oval shape and smooth or slightly lobed surface. That appearance is exactly what we found in our study (Figure 4; Appendix A). Histopathology did reveal tumors of entities other than myxoma in a third of all our patients. As mentioned, the origin of the tumor cannot be clearly assessed by other means than surgery. Only in advanced cancer cases with multiple metastasis can an intracardiac tumor be identified as such with reasonable accuracy. In those dire cases, surgery is no longer indicated. In all other circumstances, however, the tumor requires resection because of its negative local and systemic consequences and to verify the pathology. Consequently, mere thrombi are also found, which would have otherwise preferably been treated with anticoagulation alone—at least in those cases in which they are not mobile or do not alter the in- or out-flow or the valve itself. In terms of tumor pathologies, our study reflects current evidence: the majority of all tumors were myxomas, followed by 10% fibroadenomas, followed by other entities [5,6,7].

We were able to show a high survival rate with an excellent outcome and high level of satisfaction after surgery. In-hospital mortality was 2.5% (*n* = 2), other studies report a corresponding in-hospital mortality of less than 5% [19]. A single-center study from China showed a survival rate of 98.4% ± 1.6% at 5 years after surgery, very much in line with our findings [20].

The prognosis of our patients undergoing surgical resection of left atrial tumors was indeed good, corresponding with the results of the literature. There was a quick postoperative recovery, a low perioperative mortality rate, and a high level of satisfaction early as well as late.

Recurrence rate is reported to be 5% to 20% [21]. Our data, however, showed no recurrence of the resected tumors after its surgical removal, at least as far as we know from our follow-up data.

The European System for Cardiac Operative Risk Evaluation (EuroSCORE) is a cardiac risk model to predict the mortality rate for patients undergoing cardiac surgery [22]. Our data showed that women had a higher preoperative risk than men. However, that did not translate into outcome differences. It is noteworthy that the EuroSCORE system provides one risk point or 0.33% additional risk in the log for ES and 0.22% for ES II for female gender regardless of other risk factors. That alone could be the reason for the observed statistically significant differences.

## 5. Limitations

Our study has limitations, as data from a single-center study were used. Thus, the findings cannot be readily generalized. Furthermore, bias cannot be excluded in this retrospective study. Three patients were lost to follow-up. They may have also died or may have suffered from a recurrence, influencing the reported outcome presented here.

## 6. Conclusions

We could show that left atrial tumors can be operated upon with excellent early and long-term results for both entire absence of recurrence as well as high satisfaction after surgery. Patients suffering from myxoma and other left atrial tumors suspected to myxoma are predominantly female. Diagnosis may be more delayed than in men, but this is speculative. They present with a higher preoperative risk score but showed less severe accompanying coronary artery disease. Given better information and surveillance, diagnostic delay may be sufficiently addressed not only in women but in men as well, reducing the percentage of preoperative neurological complications. Aside from the aforementioned prevalence, the higher estimated risk without consequences, and lower percentage of accompanying coronary artery disease in women, no significant gender-related differences were found.

## Figures and Tables

**Figure 1 jcm-12-02075-f001:**
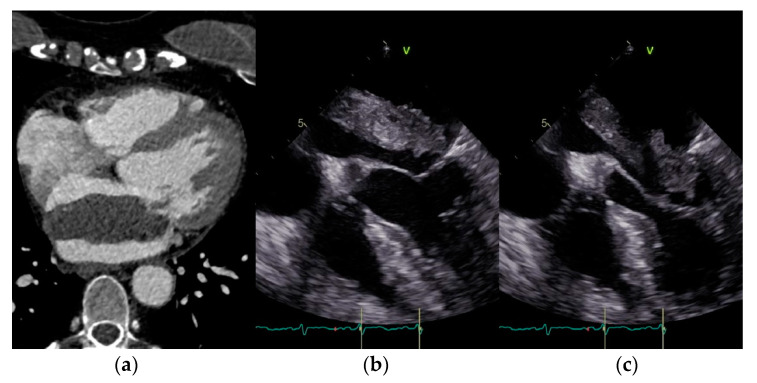
(**a**–**c**) Exemplary CT and echocardiographic findings of a left atrial myxoma attached to the interatrial septum.

**Figure 2 jcm-12-02075-f002:**
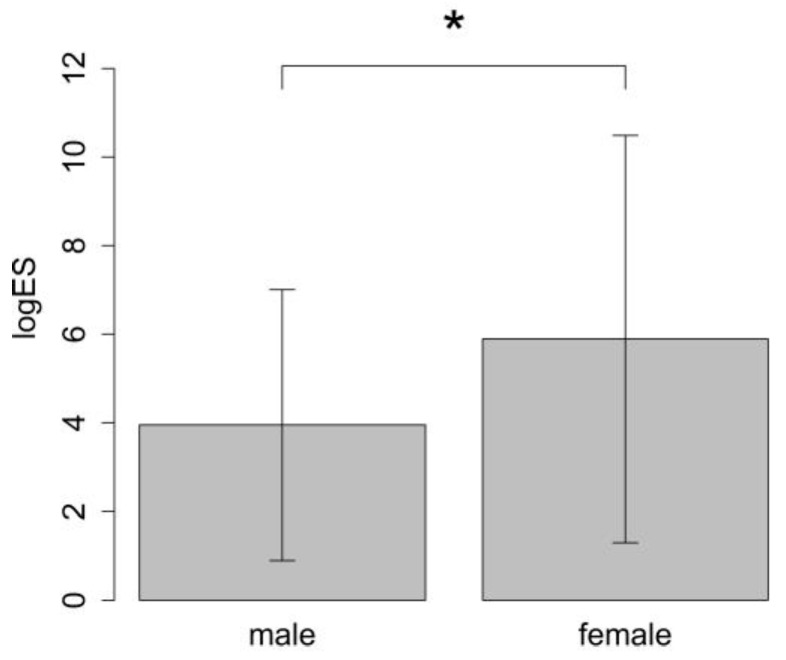
Comparison of mean LogES between female and male patients. * *p* < 0.017.

**Figure 3 jcm-12-02075-f003:**
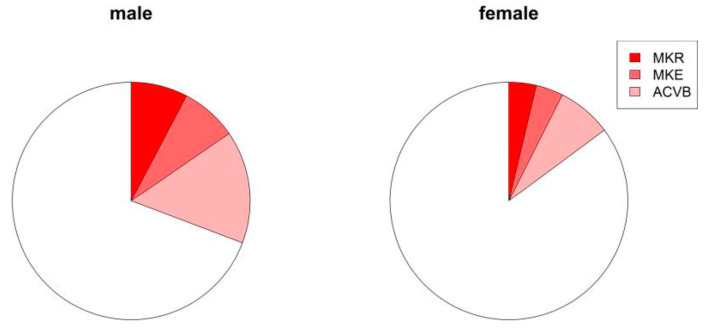
Proportion of concomitant procedures in patients with myxoma. MKR: mitral valve reconstructions, and MKE: mitral valve replacement and ACVB: coronary artery bypass graft).

**Figure 4 jcm-12-02075-f004:**
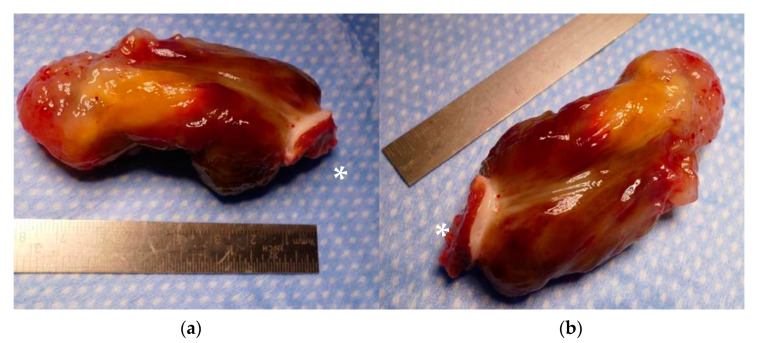
(**a**,**b**) Exemplary resected left atrial myxoma of approximately 70 mm length and 30 mm diameter. Polypoid tumor with oval shape and smooth, slightly lobulated surface. Note the resected part of the interatrial septum with septal myocardium to which the myxoma is attached (*).

**Figure 5 jcm-12-02075-f005:**
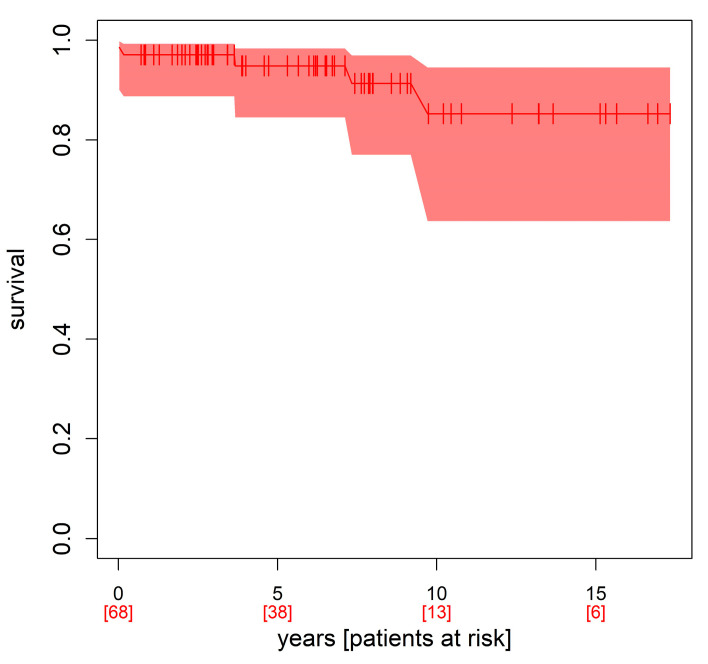
Kaplan–Meier survival curve.

**Table 1 jcm-12-02075-t001:** Basic characteristic of patients with left atrial tumors (*n* = 80, *n* female = 64, *n* male = 16).

Total (Patients) *n* = 80	Female Patients	Male Patients	*p*
Age [years]	62.76 ± 13.42	59.65 ± 15.84	0.438
BMI [kg/m^2^]	27.36 ± 6.16	27.09 ± 5.75	0.945
Euroscore	5.42 ± 2.41	4.04 ± 2.14	0.013
Logistic ES	5.89 ± 4.6	3.95 ± 3.06	0.017
ES II	2.07 ± 2.1	0.94 ± 0.45	0.043
X-clamp time [min]	41.62 ± 29.61	41.8 ± 37.8	0.6
Bypass time [min]	77.45 ± 40.86	71.73 ± 45.28	0.24
In hospital stay [d]	17.25 ± 23.52	16.92 ± 10.46	0.399
LV-EF preoperative	59.21 ± 5.1	57.67 ± 10.19	0.247
LV-EF postoperative	58.07 ± 5.99	57.5 ± 7.98	0.847
Postoperative wound-healing problems	3.85% [2]	7.69% [2]	*0.597*

BMI: body mass index; LV-EF: left ventricular ejection fraction; ES II: EuroSCORE II.

**Table 2 jcm-12-02075-t002:** Differences in symptoms between female and male patients (*n* female = 64, *n* male = 16).

Symptoms	Female Patients	Male Patients	*p*
Dyspnea	25.49% [13]	16% [4]	0.398
Atrial fibrillation	27.45% [14]	19.23% [5]	0.609
TIA	11.76% [6]	0% [0]	0.169
Stroke	19.61% [10]	12% [3]	0.526
Pulmonary diseases	15.69% [8]	24% [6]	0.573
Angina pectoris	11.76% [6]	8% [2]	1
Fever	0% [0]	8% [2]	0.105
Arterial hypertension	70.59% [36]	61.54% [16]	0.586
Diabetes mellitus	19.61% [10]	16% [4]	1
Hyperlipidemia	39.22% [20]	40% [10]	1

AP: angina pectoris; TIA: transitory ischemic attack.

**Table 3 jcm-12-02075-t003:** Concomitant coronary artery disease.

Coronary Artery Disease	Female	Male	*p*
0	78.43% [40]	68% [17]	0.481
1	7.84% [4]	4% [1]	1
2	11.76% [6]	12% [3]	1
3	1.96% [1]	16% [4]	0.038
**Coronary Artery Bypass Grafting**	**Female**	**Male**	** *p* **
0	92.59% [50]	84.62% [22]	0.427
1	3.7% [2]	0% [0]	1
2	3.7% [2]	3.85% [1]	1
3	0% [0]	11.54% [3]	0.032

**Table 4 jcm-12-02075-t004:** Histopathology (*n* = 80).

Histopathology	*n*	%	Male. *n*	Male. %	Female. *n*	Female. %	*p*
Myxoma	55	68.75	18	72	37	67.27	1
Thrombus	11	13.75	2	8	9	16.36	0.496
Fibroelastoma	8	10.00	2	8	6	10.91	1
Hamartoma	1	1.25	1	4	0	0	0.321
Thyroid gland	1	1.25	0	0	1	1.82	1
Thymus	1	1.25	0	0	1	1.82	1
Metastasis	1	1.25	1	4	0	0	0.321
Sarcoma	1	1.25	0	0	1	1.82	1
Carcinoid	1	1.25	1	4	0	0	0.321

**Table 5 jcm-12-02075-t005:** Proportion of deceased patients and patients alive today and causes of death.

Total Number of Patients	80 (100%)
Number of patients who have died	11 (13.75%)
Number of patients who were alive in 2023	69 (86.25%)
Heart disease as cause of death	3
Cancer as cause of death	1
COVID-19 as cause of death	2
Unclear cause of death	5
Could not be reached by phones	3
Follow-up patients	66 (82.5%)

**Table 6 jcm-12-02075-t006:** Level of satisfaction with perioperative treatment (*n* = 66).

*n* = 66	Very Satisfied	Satisfied	Neutral	Unsatisfied	Very Unsatisfied	Patient Reached by Phone
Level	1	2	3	4	5	
Satisfaction with perioperative treatment	57	7	2	0	0	*n* = 66
Male	15	4	0	0	0	
Female	42	3	2	0	0	

**Table 7 jcm-12-02075-t007:** Follow-up of the patients: satisfaction today (recovery and reintegration into everyday life after surgery).

*n* = 66	Very Satisfied	Satisfied	Neutral	Unsatisfied	Very Unsatisfied	Patient Reached by Phone
Male	14	4	1	0	0	*n* = 66
Female	39	6	2	0	0	

**Table 8 jcm-12-02075-t008:** Mean and median of treatment satisfaction for both sexes.

	Male Mean	Male Median	Female Mean	Female Median	*p*
Satisfaction with perioperative treatment	1.2 ± 0.41	1	1.15 ± 0.47	1	0.38
Satisfaction today	1.3 ± 0.57	1	1.22 ± 0.51	1	0.499

## Data Availability

Data is unavailable due to ethical restrictions.

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
