# Peer review of "Gender-Related Outcomes after Surgical Resection and Level of Satisfaction in Patients with Left Atrial Tumors"

_jcm, 2023, doi:10.3390/jcm12052075_

Round 1
Reviewer 1 Report
I read your paper “Gender related outcomes after surgical resection and level of of satisfaction in patients with left atrial tumors" with interest.
In this study, the authors address an important issue that has not been elucidated in the past. The gender related differences in patients undergoing cardiac surgery for different types of left atrial tumors.
In general, the article is thoroughly researched and well written. However, there are some shortcomings in the preparation and presentation of the results. In the present form, the results section is difficult to read and appears somewhat confusing. The authors are encouraged to better structure the results and clarify the issues raised.
Abstract
The abstract is too short and not very clear written, too many abbreviations, define at least Logistic ES, ESII samd. What means “late mortality”, “excellent early and late results”?
Introduction:
The introduction is moderately concise (too descriptive) but finally adequately describes the motivation of the study. It would be better to highlight more clearly why you thought about this study
Methods:
The methods section describes how the study was conducted, but without describing enough the endpoints (which is the meaning of de “lond term outcome” for this study), the follow-up visits, which parameters were assessed. Also, statistical analysis is strictly descriptive, for this analysis is not necessary SPSS. The statistical models used to determine some correlations, independent predictors should be described and used (for example, corellation coefficients, statistical methods, ANOVA square sums, F values, or post hoc results, correlation analysis). At least, the input and output values should be clarified. Also, it is necessary a discussion about the level of satisfaction, how did you calculate/evaluate it? Figure 3 has no usefulness….and it is not representative, it is described into text…
The authors are encouraged to go into a bit more detail with statistical analysis, not only a simple description. To give a conclusion, this analysis is very important, otherwise it is only an observational study.
The description of the patient evaluation is very poor, difficult to understand which parameters were followed preoperatively and postoperatively…
Results:
It is difficult to see through the presentation of the results. It is a long description which is difficult to follow. I would like to list some important points.
Table 1 and Table 2 presented basic characteristics of the patients and differences in symptoms between and there is a mixture between surgical characteristics, Euroscore, echographic parameters (LVEF), comorbidities (stroke, pulmonary disease, diabetes, arterial hypertension) and symptoms.The presentation is not enough structured, it is difficult to read.
In order to be easier to understand, the main demographic,clinical and echographic characteristics of the two subgroups would be better presented in a table (mean age, BMI, symptoms, atrial fibrillation - % patients and number with AF, thrombus, echo contrast, LEEF), comorbidities (diabetes…samd….). In the second table you should present characteristics related to surgical procedure, surgical risk and postoperative evolution. If you want to empashize the differences between male and female and if and if women are more exposed to postoperative complications you should perform a regression analysis.
Figure 2 is not well chosen to represent the differences in LogES between male and female. Also, Figure 3 is not well chosen to represent concomitant procedures, maybe a Pie chart is better…
Figure 4 is useless, the paper is not a case report. Reformulate the title for Table 4.
Please describe how did you measure the “ level of satisfaction”
The follow up is insufficient described, which parameters were followed at each visit, which are the endpoints…. Please elaborate on the follow-up visits in the results and discussion.
The results consists mainly in a too long description, comparison of means, difficult to follow. Please structure them better
It is generally difficult to follow the follow up data. Maybe if you structured them in a table it would be easier…
Discussion:
The discussion is mainly based on the embedding of the left atrium tumors in the existing literature. . Discussions are somewhat confusing and not enough elaborated so that it is not easy to get the point. A detailed comparison with other studies would be welcome
I didn’t understand the meaning of Kaplan-Meier-Survival Curve in the Discussion. A figure in the Discussion section it makes no sense…. On the other hand, you stated in the statistical methodology part that you used survival curves, but I did not find them in the text
The reference list is very extensive, and not very relevant to the topic. The follow-up results that had been collected should also be included in the discussion
However, the authors should a little more focus on the clinical conclusions. They have found that that women present a higher preoperative risk score but showed less severe accompanying coronary artery disease. Is it just another question or did they encourage larger studies or to provide a patient selection strategy for screening for cardiac tumors according to their results? Should it have an impact on the current guidelines?
In view of the very poor statistical methods used, I would consider it advisable to consult a statistical reviewer. This might be helpful to get the presentation of the results more structured.
Minor English language improvement should be performed.
Reviewer 2 Report
Comments to the Author
The manuscript entitled: “Gender-related outcomes after surgical resection and level of satisfaction in patients with left atrial tumors” described gender differences in the frequency of left atrial mass development and postoperative outcomes. Although the authors showed interesting findings, I would like to raise the following concerns.
Q1.
It is advisable to incorporate the distinction between male and female participants in each table for heightened comprehensibility. For instance, this could be represented as "Females (n=64).
Q2.
Kindly furnish supplementary information regarding the inadequate elaboration on the basis for emphasizing gender-based outcomes and patient satisfaction with surgical treatment for left atrial tumors in this study.
Q3.
Approximately 5% of left atrial myxomas are hereditary, and these cases tend to manifest in young males and are often multiple and recurrent in nature. Were these types of cases taken into consideration in the study?
Q4.
What are the predictive factors that can be correlated with the long-term prognosis following surgery for a left atrial mass, such as age, histological type, and surgical technique? Is concurrently implementing valve replacement or coronary artery bypass surgery considered a negative prognostic indicator?
Round 2
Reviewer 1 Report
The work is clearly improved and can be published